# Study of the influencing factors of the liquid $CO_2$ phase change fracturing effect in coal seams

**Jinzhang Jia**[1,2], **Dongming Wang**[1,2], **Bin Li**[1,2]*, **Xiuyuan Tian**[1,2]

**1** College of Safety Science and Engineering, Liaoning Technical University, Fuxin, Liaoning, China, **2** Key Laboratory of Mine Power Disaster and Prevention of Ministry of Education, Huludao, Liaoning, China

* 30727899@qq.com

**Data Availability Statement:** All relevant data are within the manuscript.

**Funding:** This research was supported by the National Natural Science Foundation of China (No. 51374121) and funded by Liaoning Distinguished

## Abstract

To study the influence of different factors on the cracking effect of the liquid $CO_2$ phase transition, the mechanics of coal rock crack extension based on liquid $CO_2$ phase change blast loading were studied. Through the application of simulation software to analyze the influence of coal seam physical parameters (in situ stress, gas pressure, modulus of elasticity and strength of coal) and blasting parameters (fracturing pore size and peak pressure of detonation)on the effect of liquid $CO_2$ phase change cracking, the simulation results showed that the cracking effect of liquid $CO_2$ phase change was positively correlated with the changes in gas pressure, elastic modulus, fracture hole diameter and peak vent pressure, negatively correlated with the variation in situ stress and compressive strength, and nearly independent of the tensile strength. In addition, by using Gray correlation analysis to analyze the influence degree of six main factors on the cracking effect, the calculation results showed that the effect of blasting parameters was greater than that of physical parameters. The main controlling factor that affected the blasting effect was the peak pressure of blasting release. By conducting comparative engineering trials with different blasting parameters, the test results showed that the crack effect of the coal seam was positively correlated with the change in fracture hole diameter and peak venting pressure, which was consistent with the results obtained from the simulation. The experimental results and simulation results for the effective radius of coal seam fracturing were basically consistent, with the error between the two types of results falling below 10%. Therefore, the reliability of the blasting numerical model was verified. In summary, the research results provide theoretical guidance for applying and promoting liquid $CO_2$ fracturing technology in coal mines.

## 1. Introduction

With an increasing mining depth, gas hazards become increasingly severe [1, 2]. To increase the rate of gas extraction and ensure safe coal mining, therefore, coal seam penetration enhancement technology is widely used. At present, liquid $CO_2$ phase change fracturing technology is a relatively safe and reliable method for increasing permeability; it not only increases

Professor (551710007007), funded project of the Liaoning Million Talents Cover Letter project (2019-45-15), and the Natural Science Foundation of Liaoning Province (2019-MS-162). The funders had no role in study design, data collection and analysis, decision to publish, or preparation of the manuscript.

**Competing interests:** The authors have declared that no competing interests exist.

the permeability of the coal seam and improves the efficiency of gas extraction, but also effectively prevents the spread of gas and prevents coal and gas protrusion [3].

In recent years, numerous researchers have conducted extensive research on liquid $CO_2$ phase change fracturing technology. Singh [4] stated that liquid $CO_2$ phase change rock breaking technology was not constrained by explosive blasting and could work efficiently and continuously, making it suitable for rock mining. Bennour et al. [5] studied the fracture extension characteristics of shale core water pressure, oil pressure and liquid $CO_2$ phase change fracturing and showed that the core was destroyed, showing type II cracks. The crack was relatively wide, and many branch cracks were generated along the main crack under liquid $CO_2$ fracturing conditions. Ishida et al. [6] found that it could form more extensive and complex macrodestructive cracks to use liquid $CO_2$ instead of water for reservoir fracturing. Luo et al. [7] investigated the rheological properties of liquid $CO_2$ during fracturing and found that the viscosity of $CO_2$ was positively correlated with temperature and shear rate. Yang et al. [8] used thermodynamic equations to evaluate liquid $CO_2$ phase change burst energy, established a model for high-pressure gas explosions in coal based on the SPH algorithm, and the fracture zone and crack zone range and criteria for crack propagation under high-pressure gas were calculated. Hu et al. [9] investigated the phase change explosion process of liquid $CO_2$ by numerical simulation, and designed an on-site fracture drilling arrangement. The results show that coal permeability and gas extraction efficiency increase substantially after liquid $CO_2$ blasting, and the gas extraction from the borehole was 1.8- to 8-fold the gas extraction from the original borehole after blasting. Bai et al. [10] developed an experimental device for liquid $CO_2$ phase change jet coal rock fracturing. Based on the principle of liquid $CO_2$ phase change fracturing technology, they conducted a study of the decay law of liquid $CO_2$ phase change jet pressure with time and liquid $CO_2$ phase change jet pressure with fractured coal rock macro and micro damage law. Chen et al. [11] studied by experimental and numerical simulations the application of the liquid $CO_2$ phase change fracturing technique to changing the permeability of coal seams, and the results showed that liquid $CO_2$ phase change fracturing substantially improved the permeability of coal seams. Pan et al. [12] proposed a new method for discontinuous rock fracturing simulation and applied it to liquid $CO_2$ fracturing to demonstrate its effectiveness in multiphase flow fracturing simulations.

The above research results have laid a solid foundation for research on the phase change fracturing technology of liquid $CO_2$ in coal seams. However, because of the complicated conditions under which underground coal seams occur, many factors affect the effect of blasting fracturing, and the degree of influence of these factors on the fracture-causing effect is unclear. As a result, this technique may cause unsatisfactory blasting results or higher blasting costs in the application process. This paper investigates by simulation software of ANSYS/LS-DYNA the degree of influence of each factor on the phase change fracturing effect of liquid $CO_2$, to analyze the influence of coal seam physical parameters(in-situ stress、gas pressure、modulus of elasticity and strength of coal) and blasting parameters (in situ stress, gas pressure, modulus of elasticity and strength of coal) and blasting parameters (fracturing pore size and peak pressure of detonation) on the effect of liquid $CO_2$ phase change cracking. The main control factor among multiple influencing factors was identified by gray correlation theory. In addition, the effect of different blasting parameters on the fracturing effect of coal seams was studied by conducting field tests, and the reliability of the blasting numerical model was verified. The research results have important application value for optimizing on-site drilling layouts and guiding technical construction.

## 2. Principle of liquid CO$_2$ phase change cracking technology and the mechanical mechanism of crack extension

### 2.1 Principle of liquid CO$_2$ phase change fracturing technology

The principle of liquid CO$_2$ phase change fracturing technology [13] is to use a booster pump to pressurize and fill the liquid CO$_2$ into a storage tube with a built-in electrically heated activator and a constant pressure relief plate, until the CO$_2$ gas in the gas storage tube changes to a high-pressure critical state. Through manual remote operation, the built-in electric heating activator is activated by controlling the microcurrent to make the liquid CO$_2$ in the gas storage cavity instantaneously boil and expand, achieving the ultimate shear strength of the fixed pressure relief sheet. A large amount of high-pressure gaseous CO$_2$ came out of the valve body and acted directly on the coal wall of the borehole, after the deflagration piece was destroyed, as shown in Fig 1. High-pressure CO$_2$ gas caused the coal body to form a macroscopic fracture zone and numerous microscopic crack networks; thus, it has promoted the free resolution of coal seam gas and realized the fracturing and increased penetration of the low-permeability coal seam.

### 2.2 Mechanism of crack extension by phase change cracking of liquid CO$_2$ in a coal seam

**2.2.1 Analysis of the phase change fracturing mechanism of liquid CO$_2$.** The mechanical model of the liquid CO$_2$ phase change fracturing coal body is shown in Fig 2. When a liquid CO$_2$ fissioner detonates, high-pressure gas rushes out from the gas hole to form a shock wave on the coal body to produce radial compression, and its strength far exceeds the compressive strength of the coal body, causing the coal body to break. As the shock wave decays, reflection occurs after the stress wave propagates to the free surface, and the pressure drops rapidly from positive to negative values to become a tensile wave, causing tangential tensile cracks within the coal. After the formation of initial cracks in the coal body, stress concentration is generated at the tip of the initial crack under the combined action of gas pressure, ground stress and burst gas pressure, and the initial crack is further expanded, followed by the formation of a larger-scale crack network [14, 15].

**2.2.2 TNT-equivalent conversion of liquid CO$_2$ phase change fracturing.** The liquid CO$_2$ was rapidly converted from liquid to gas after being heated by the heating device, the expansion of the gas medium inside the fissioner was an adiabatic process, and the energy released by the blast was equivalent to the work done by the expansion of the gas, as shown in the following equation [16]:

$$E_g = \frac{P_0 V_L}{K-1}\left[1 - \left(\frac{P^{(K-1)/K}}{P_0}\right)\right] \times 10^3 \tag{1}$$

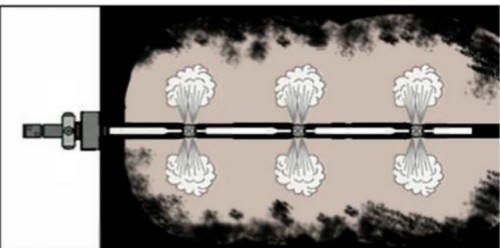

**Fig 1. Schematic diagram of phase change fracturing of liquid CO$_2$.**

where $E_g$ is the gas burst energy in kJ; $P_0$ is the peak pressure of the fission device release in MPa; P is the standard atmospheric pressure, taken as 0.101 MPa; $V_L$ is the fission volume in m$^3$; K is the adiabatic index of the medium, taken as 1.295 for liquid CO$_2$.

The approximate TNT equivalent of the energy released from the liquid CO$_2$ blast [16] was calculated using the following equation:

$$W_{\mathrm{TNT}} = \frac{E_g}{Q_{\mathrm{TNT}}} \tag{2}$$

where $W_{\mathrm{TNT}}$ is the release equivalent of TNT explosives in kg, and $Q_{\mathrm{TNT}}$ is the explosion energy of the 1 kg TNT explosive, which is 4250 kJ/kg.

**2.2.3 Liquid CO$_2$ phase change gas explosion load process analysis.**   Airburst loading was the main driver of liquid CO$_2$ phase change fracturing of coal bodies, and the airburst load was not constant during the blasting process [17]. According to the size of the gas explosion load and the extension of the propagation distance, the explosive load process can be divided into three stages. When $L$ was more than 0 and less than $L_1$, it was the stage of explosion shock wave action, and the peak load $P_1$ was the impact force when the burst gas collided with the gun hole wall. When $L$ was more than $L_1$ and less than $L_2$, it was the stress wave action stage, and the peak load $P_2$ was the boundary stress of the coal crushing zone. When $L$ was more than $L_2$ and less than $L_3$, it was the burst of the gas pressure action stage, and the peak load $P_3$ was the quasi-static pressure when the burst gas filled the gun hole. The peak blast load in each phase is given by the following equation.

$$P_{\max} = \begin{cases} P_1 & 0 < L < L_1 \\ P_2 & L_1 < L < L_2 \\ P_3 & L_2 < L < L_3 \end{cases} \tag{3}$$

The quasi-static pressure of the burst gas can be calculated according to the isentropic expansion process, assuming that the burst process is brief, and the expansion of the burst gas is an adiabatic process [17]. $P_3$ is expressed as follows:

$$P_3 = P_0 \cdot \left[ \frac{\rho_0 D_0^2}{2(k+1)P_0} \right]^{\gamma/k} \cdot \left( \frac{V_L}{V_H} \right)^{\gamma} \tag{4}$$

where $\rho_0$ is the density of liquid CO$_2$ at a common temperature in kg/m$^3$, $D_0$ is the fissioner burst gas explosion velocity in m/s, $k$ is the adiabatic isentropic index of the burst gas, $\gamma$ is the adiabatic index of the burst gas, and $V_H$ is the volume of the fracturing hole in m$^3$.

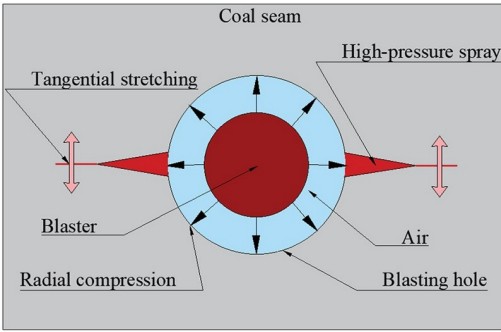

**Fig 2. Mechanical model of fracture by liquid CO$_2$ blasting in a coal body.**

Under engineering blasting conditions, the explosive gas body expansion filled with gun holes, the impact pressure generated by the collision with the hole wall increased substantially, and the peak load $P_1$ acting on the wall of the gun hole can be expressed by the following equation:

$$P_1 = n \cdot P_0 \cdot \left[ \frac{\rho_0 D_0^2}{2(k+1)P_0} \right]^{\gamma/k} \cdot \left( \frac{V_L}{V_H} \right)^{\gamma} \tag{5}$$

where $n$ is the pressure increase coefficient, and $n$ is taken from 8 to 10.

The liquid $CO_2$ gas explosion shock wave propagated within the coal and rapidly decayed into a stress wave. The stress at the boundary of the crush zone was the peak stress wave load $P_2$, given by the following equation [17]:

$$P_2 = \sigma_{cd} = \sigma_c \cdot \sqrt[3]{\varepsilon} \tag{6}$$

where $\sigma_{cd}$ is the dynamic compressive strength of the coal body in MPa, $\sigma_c$ is the static compressive strength of the coal body in MPa, and $\varepsilon$ is the loading strain rate.

In summary, the peak liquid $CO_2$ gas explosion load at different stages is given in Eq (7).

$$P_{\max} = \begin{cases} P_1 = n \cdot P_0 \cdot \left[ \dfrac{\rho_0 D_0^2}{2(k+1)P_0} \right]^{\gamma/k} \cdot \left( \dfrac{V_L}{V_H} \right)^{\gamma} & 0 < L < L_1 \\[2mm] P_2 = \sigma_{cd} = \sigma_c \cdot \sqrt[3]{\varepsilon} & L_1 < L < L_2 \\[2mm] P_3 = P_0 \cdot \left[ \dfrac{\rho_0 D_0^2}{2(k+1)P_0} \right]^{\gamma/k} \cdot \left( \dfrac{V_L}{V_H} \right)^{\gamma} & L_2 < L < L_3 \end{cases} \tag{7}$$

**2.2.4 Mechanism of coal rock crack generation under liquid $CO_2$ phase change airburst loading.** During coal seam fracturing, a shock wave was produced by a liquid $CO_2$ phase change gas explosion, which spread outward with the blasthole as the center and rapidly decayed into stress waves. Stress waves continued to decay as they propagated. The radial stress generated by the gas explosion shock wave and stress wave acting on the coal can be expressed as the following formula [18].

$$\sigma_r = \frac{P_{\max}}{\bar{r}^{\alpha}} = \begin{cases} \dfrac{P_1}{\bar{r}^{\alpha}} & 0 < L < L_1 \\[2mm] \dfrac{P_2}{\bar{r}^{\alpha}} & L_1 < L < L_2 \end{cases} \tag{8}$$

where $\bar{r}$ is the contrast distance, $\bar{r} = r_i / r_b$, $r_i$ is the distance from any point to the center of the blasthole in m, $r_b$ is the drilling radius in m, $\alpha$ is the pressure attenuation coefficient, the shock wave phase equation is $\alpha = 2 + \mu/(1-\mu)$, the stress wave phase equation is $\alpha = 2 - \mu/(1-\mu)$, and $\mu$ is Poisson's ratio.

The tangential stress formula of the explosion shock wave and stress wave acting on coal is as follows [18].

$$\begin{cases} \sigma_\theta = -b\sigma_r \\ b = \mu/1 - \mu \end{cases} \tag{9}$$

where $b$ is the lateral stress coefficient. Since the wave velocity is relatively easy to measure, the dynamic Poisson's ratio of coal rock can be expressed in terms of the wave velocity as follows

[18].

$$\mu = \frac{c_p^2 - 2c_s^2}{2(c_p^2 - c_s^2)} \tag{10}$$

where $c_p$ is the longitudinal wave speed in m/s, and $c_s$ is the transverse wave speed in m/s.

Liquid $CO_2$ airbursts in coal bodies can be considered a plane strain problem [19], and the axial stress can be expressed by the following equation.

$$\sigma_z = \mu(\sigma_r + \sigma_\theta) = \mu(1 - b)\sigma_r \tag{11}$$

where $\sigma_z$, $\sigma_r$, and $\sigma_\theta$ are the axial, radial, and tangential stresses at any point in the coal rock in MPa, respectively.

Under the blast impact load, the stress intensity at any point in the coal rock can be expressed by the radial stress, tangential stress and axial stress [17].

$$\sigma_i = \frac{1}{\sqrt{2}} \left[ (\sigma_r - \sigma_\theta)^2 + (\sigma_\theta - \sigma_z)^2 + (\sigma_z - \sigma_r)^2 \right]^{1/2} \tag{12}$$

Substitute Eqs (9), (10) and (11) into Eq (12)

$$\sigma_i = \frac{1}{\sqrt{2}} \sigma_r \left[ (1 + b)^2 + 2\mu(1 - b)^2(1 - b) + (1 + b^2) \right]^{1/2} \tag{13}$$

Under airburst impact loading, deformation damage of the coal body was dominated by compression damage and tensile damage [17]. When the effective stress strength was greater than the dynamic compressive strength of the coal body, the coal body was compressed and destroyed, forming a crushed area. When the effective stress strength was greater than the dynamic tensile strength of the coal body, the coal body was stretched and broken, forming a cracked area. This behavior can be expressed by the following inequalities [17].

$$\begin{cases} \sigma_i \geq \sigma_{cd} & \text{(Crushed area)} \\ \sigma_i \geq \sigma_{td} & \text{(Cracked area)} \end{cases} \tag{14}$$

where $\sigma_{cd}$ is the dynamic compressive strength of the coal body, and $\sigma_{td}$ is the dynamic tensile strength of the coal body.

**2.2.5 Mechanism of coal rock crack extension under liquid CO₂ phase change airburst loading.** Coal is a quasi-brittle material. Under the action of liquid $CO_2$ blasting load, the coal structure was usually affected by the combined stress field, and there was a fracture process zone (FPZ) at the crack tip, which had a greater impact on crack propagation. According to literature [17, 20, 21], the cracks formed at this stage were mainly type I-II compound cracks, and the stress component of the crack tip region in the oblique cross section in the polar coordinate system can be expressed by the following equation.

$$\begin{cases} \sigma'_r = \frac{1}{2\sqrt{2\pi r_i}} \left[ K_I(3 - \cos\theta)\cos\frac{\theta}{2} + K_{II}(3\cos\theta - 1)\sin\frac{\theta}{2} \right] \\ \sigma'_\theta = \frac{1}{2\sqrt{2\pi r_i}} \cos\theta \left[ K_I\cos^2\frac{\theta}{2} - \frac{3}{2}K_{II}\sin\frac{\theta}{2} \right] \\ \tau'_{r\theta} = \frac{1}{2\sqrt{2\pi r_i}} \cos\frac{\theta}{2} \left[ K_I\sin\theta + K_{II}(3\cos\theta - 1) \right] \end{cases} \tag{15}$$

where $\theta$ is the angle between the normal direction of an arbitrary oblique section at the tip

point of the coal rock and the direction of the original crack (°), and $K_I$ and $K_{II}$ are the stress intensity factors of the crack tip under the combined action of far-field stress (in situ stress), explosion gas pressure and gas pressure. According to the superposition principle [21], the stress intensity factors can be expressed as follows:

$$\begin{cases} K_{\mathrm{I}} = -\dfrac{1}{2}\sqrt{\pi a}\left[(\sigma_1 + \sigma_3) - (\sigma_1 - \sigma_3)\cos 2\beta\right] + \dfrac{4}{1-D}P_3\left(\dfrac{\pi}{2}-1\right)\sqrt{\dfrac{a}{\pi}} + P_g\sqrt{\pi a} \\ K_{\mathrm{II}} = -\dfrac{1}{2}\sqrt{\pi}(\sigma_1 - \sigma_3)\sin 2\beta \end{cases} \quad (16)$$

where $a$ is the crack length in m, $\sigma_1$ and $\sigma_3$ are the far-field principal stresses in MPa, $\beta$ is the angle between the crack and the maximum principal stress in the far field (°), and $P_g$ is the average gas pressure in the crack in MPa.

Based on basic assumption 1 of maximum circumferential stress theory [17], the crack expansion direction is the angle corresponding to the maximum value of the circumferential positive stress, and the fracture criterion for composite cracks can be obtained.

$$(\sigma'_\theta)_{\max} > (\sigma'_\theta)_c \quad (17)$$

where $(\sigma'_\theta)_{\max}$ is the peak circumferential tensile stress in the coal unit in MPa, and $(\sigma'_\theta)_c$ is the critical value of the maximum circumferential stress in the coal unit in MPa.

The critical value of the maximum circumferential stress in the coal unit can be determined by the fracture toughness $K_{\mathrm{IC}}$ of type I cracks [21]. Since type I cracks always expand in the direction of the original crack, the cracking direction angle $\theta$ is equal to 0. $K_{\mathrm{II}} = 0$, $\theta = 0$, and $K_{\mathrm{I}} = K_{\mathrm{IC}}$ are brought into Eq (15) to obtain the critical value of the maximum circumferential stress.

$$(\sigma_\theta)_c = \dfrac{K_{\mathrm{IC}}}{\sqrt{2\pi r_i}} \quad (18)$$

During coal rock blasting, the peak circumferential tensile stress at the tip of the type I-II composite crack can be expressed by the following equation [17]:

$$(\sigma_\theta)_{\max} = \dfrac{1}{\sqrt{2\pi r_i}}\cos\dfrac{\theta}{2}\left(K_{\mathrm{I}}\cos^2\dfrac{\theta}{2} - \dfrac{3}{2}K_{\mathrm{II}}\sin\theta\right) \quad (19)$$

Therefore, under the combined effect of explosive gas, gas pressure and in situ stress, the crack initiation condition for crack expansion can be expressed by the following equation:

$$\cos\dfrac{\theta}{2}\left(K_{\mathrm{I}}\cos^2\dfrac{\theta}{2} - \dfrac{3}{2}K_{\mathrm{II}}\sin\theta\right) > K_{IC}(\le K_{IC}) \quad (20)$$

According to the above analysis, the destruction of the coal body was a rather complex kinetic process under the action of the blast impact load. After liquid CO$_2$ blasting, the combined effect of explosion shock waves, stress waves and explosive gas changed the original stress state of the coal body. The fracturing process of the coal body was influenced by the combination of in situ stress, gas pressure, explosion load and other factors, especially the dynamic effect of drilling and blasting loads, and the analysis difficulty of coal crack extension was further increased. Therefore, the visualization of blasting effects can be studied with the help of simulation software.

## 3. Numerical simulation analysis of the factors influencing the effect of phase change fracturing of liquid CO$_2$

Based on the mechanical mechanism of coal rock crack extension under the action of a liquid CO$_2$ phase change burst load, to more accurately study the effect of phase change fracturing of liquid CO$_2$, the effects of the main physical parameters (in situ stress, gas pressure, modulus of elasticity, coal body strength) of the coal body and the main blasting parameters (drill hole diameter, energy relief piece pressure peak) on the phase change fracturing effect of liquid CO$_2$ were analyzed by using simulation software of ANSYS/LS-DYNA.

### 3.1 Model and parameters

In this paper, based on the actual conditions of the test working face of the Ma bao coal mine and combined with the relevant coal and rock parameters in the literature [22, 23], a three-dimensional numerical analysis model of coal seam liquid CO$_2$ phase change fracturing was constructed. The model was composed of three parts: cracker, air and coal, using fluid-solid coupling algorithm. The geometric model size is 10 m×10 m×0.1 m, as shown in Fig 3. The model is meshed into 126,626 units, the Z-axis direction is set on the front and rear surfaces of the constraint model, the Y-axis direction is set at the upper and lower boundaries for constraint, and the X-axis direction is set at the left and right boundaries for constraint. To simulate the blasting process in an infinite coal body and eliminate the effect of reflection superposition of stress waves at the boundary of the constructed model on crack extension, the boundary surface is set to a reflection-free boundary. The physical mechanics parameters of the coal seam are shown in Table 1.

TNT equivalent conversion is based on liquid CO$_2$ phase change cracking, and the explosion relief pressure $P_0$ can be expressed by the JWL state equation [23]:

$$P_0 = A\left(1 - \frac{\omega}{R_1 V}\right)e^{-R_1 V} + B\left(1 - \frac{\omega}{R_2 V}\right)e^{-R_2 V} + \frac{\omega E_0}{V} \tag{21}$$

where $E_0$ is the initial internal energy in GPa, and $V$ is the specific volume in m$^3$. A, B, $R_1$, $R_2$, and $\omega$ are material constants associated with explosives, which can be obtained by fitting the TNT density $\rho$, burst velocity $D_0$ and adiabatic coefficient $\gamma$. The explosive parameters and JWL equation of state parameters are shown in Table 2.

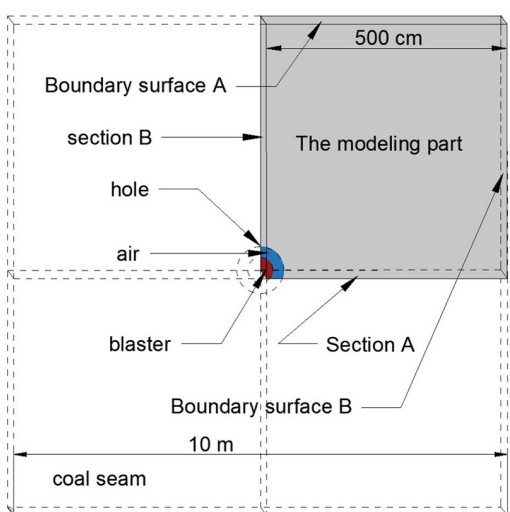

**Fig 3. Numerical simulation model of liquid CO$_2$ blasting in a coal seam.**

Table 1. Parameters related to coal rock materials.

| parameter | value | parameter | value |
|---|---|---|---|
| Mass density/(kg/m$^3$) | 1350 | Reference compressive strain rate | 3e$^{-5}$ |
| Failure surface parameter | 1.7 | Reference tensile strain rate | 3e$^{-6}$ |
| Eroding plastic strain | 1.2 | Break compressive strain rate | 3e$^{25}$ |
| Poisson ratio | 0.3 | Break tensile strain rate | 3e$^{25}$ |
| Initial porosity | 2.2 | Pressure influence on plastic flow in tension | 0.001 |
| Porosity exponent | 6.5 | Damage parameter-D1 | 0.02 |
| Lode angle dependence factor | 0.72 | Damage parameter-D2 | 1 |
| Tensile yield surface parameter | 0.4 | Minimum damaged residual strain | 0.01 |
| Compressive yield surface parameter | 0.85 | Relative shear strength | 0.07 |

## 3.2 Analysis of simulation results

To more intuitively express the fracturing effect of the coal body under the action of liquid CO$_2$ phase change fracturing, in numerical simulation, two methods are usually used to quantitatively analyze the cracking effect [24]: the first is to count the effective radius of blasting fracturing (R), and the evaluation index [25] of the effective radius of fracturing is based on the three parts of the coal body after blasting, namely, the crushed area, the cracked area and the cracked extension area, as shown in Fig 4. The second is the statistical coal body fracture degree (M), which is an index to evaluate the degree of coal fracture, which is the ratio of the volume of the fractured coal body to the total volume of the coal body. In the simulation analysis, it is expressed as the ratio of the volume of the failed element to the volume of all elements. According to the relevant literature [26–28], the range of parameter values for the different influencing factors was described, and the values of the parameters in this paper are shown in Table 2. During the simulation, in addition to changing the relevant parameters of the studied factors, other parameters were set to the first set of values in Table 3 to ensure the reliability of the conclusions.

**3.2.1 Analysis of the effect of *in situ* stress on the effective range of cracking.** Fig 5A–5D show the simulation results of the phase change fracturing of liquid CO$_2$ in a coal seam with in situ stresses of 15, 18, 21 and 24 MPa, respectively. The circles from inside to outside indicate the aperture, crush zone, crack dense zone, and crack extension zone, respectively. The effective radius of fracture was measured to be approximately 357, 335, 317, and 305 cm. The ratio of the volume of failed units to the volume of all units was calculated to obtain coal body fracture degrees of approximately 6.35, 5.96, 5.63, and 5.22%.

The simulation results show that the effective radius and fracture degree of the liquid CO$_2$ phase change fracture gradually decreased with increasing in situ stress. The reason was that the hoop tensile stress generated by the explosion load of liquid CO$_2$ gas could cause radial cracks in the coal seam, but the in situ stress in the coal seam would produce hoop compressive stress, which would inhibit the tension effect of the explosion, making the coal body less prone to rupture damage and formation of initial cracks. In addition, the higher in situ stress in the coal seam would reduce the stress intensity factor at the crack tip of the coal body under the action of a quasi-static stress field, which was not conducive to crack expansion. Therefore, during the actual underground construction process, the space and location of blast holes in

Table 2. Explosives equation of state parameters.

| $\rho$/(kg/m$^3$) | $D_0$/(m/s) | $\gamma$ | $A$ | $B$ | $R_1$ | $R_2$ | $\omega$ | $E$/(GPa) |
|---|---|---|---|---|---|---|---|---|
| 0.9 | 3600 | 1.33 | 332 | 0.752 | 4.07 | 0.98 | 0.23 | 1.0 |

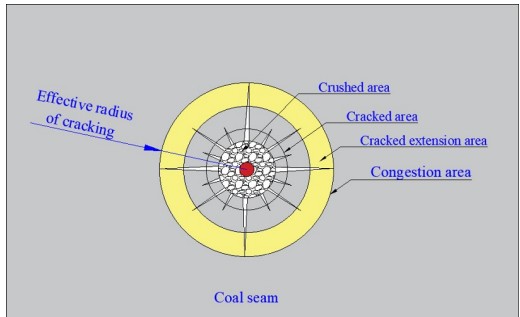

**Fig 4. Schematic diagram of the effective radius of phase change fracturing of liquid CO$_2$.**

the coal seam should be reasonably arranged according to the value of in situ stress to achieve the best blast fracturing effect.

**3.2.2 Analysis of the effect of gas pressure on the effective range of fracturing.** Fig 6A–6D show the simulation results of the phase change fracturing of liquid CO$_2$ in a coal seam with gas pressures of 2, 4, 6 and 8 MPa, respectively. The measured effective radii of fracture were approximately 357, 368, 379, and 391 cm, and the coal fracture degrees were approximately 6.35, 6.54, 6.82, and 7.04%, respectively.

The simulation results show that the effective radius and fracture degree of liquid CO$_2$ phase change fracturing gradually increased with increasing gas pressure. The reason was that liquid CO$_2$ phase change cracking blasting was conducted in the solid-fluid coupling medium of coal and gas, the effect of gas pressure caused cracks in the coal to expand and open, and stress concentration was generated within the coal skeleton at the crack tip, which put the crack extension in dynamic equilibrium. Under the action of the explosion load of the liquid CO$_2$ gas, the gas pressure increased instantaneously; under the combined action of the explosion gas and the gas pressure, the cracks began to propagate in the coal skeleton. The higher the gas pressure was, the greater the stored energy and the stronger the response to explosive loads, the easier the coal body ruptured, and the easier it was for cracks to expand. In addition, when the gas pressure within the coal seam was high, the effective stress of the coal body itself was reduced. After blasting, the stress to be overcome for crack expansion was reduced, so the effective fracture was increased. It can be concluded that the coal seam gas pressure to a certain degree is conducive to the expansion of airburst cracking.

**3.2.3 Analysis of the effect of elastic modulus on the effective range of cracking.** Fig 7A–7D show the simulation results of the phase change fracture of liquid CO$_2$ in the coal seam with elastic moduli of 6.5, 4.5, 2.5 and 0.5 GPa, respectively. The effective radius of fracture was measured as 357, 339, 324, 311 cm, and the fracture degree of the coal body was 6.35, 6.03, 5.86, and 5.75%, respectively.

**Table 3. Parameter values of each factor.**

| Influencing Factors | Parameter Value | | | |
|---|---|---|---|---|
| *In situ* stress/MPa | 15 | 18 | 21 | 24 |
| Gas pressure/MPa | 2 | 4 | 6 | 8 |
| Modulus of elasticity/GPa | 6.5 | 4.5 | 2.5 | 0.5 |
| Tensile strength of coal body/MPa | 0.2 | 0.5 | 0.8 | 1.1 |
| Compressive strength of coal body/MPa | 20 | 16 | 12 | 8 |
| Pore size/mm | 133 | 113 | 94 | 75 |
| Peak pressure of energy relief plate/MPA | 180 | 210 | 240 | 270 |

R=357cm M=6.35%  R=335cm M=5.96%  R=317cm M=5.63%  R=308cm M=5.22%

(a)　　　　　(b)　　　　　(c)　　　　　(d)

**Fig 5. Crack distribution under different ground stresses.** (a)*In situ* stress 15 MPa.(b)*In situ* stress 18 MPa.(c)*In situ* stress 21 MPa.(d)*In situ* stress 24 MPa.

The simulation results show that the effective radius of cracking and the degree of rupture decrease with decreasing elastic modulus. The reason was that the smaller the modulus of elasticity of the coal body was, the lower the stiffness of the coal body, and under the action of an explosive gas shock wave, the larger the area around the fracture hole to form a smash circle was, the more severe the damage to the coal body, and the more obvious the effect of energy absorption of explosions. A large concentration of energy consumption in the smash circle resulted in severe energy damage, which was not conducive to the further expansion of the crack. In addition, with the increase in the fracture degree of rock mass, the ability to resist stress waves became increasingly strong, and the peak value of stress waves decayed faster, which caused the number and length of the initial radial cracks generated by the stress wave action to decrease, and the effect of the quasi-static action of the blast-generated gas with a low peak at the later stage was substantially weakened, leading to a reduction in the range of the crack zone finally generated by blasting.

**3.2.4 Analysis of the effect of coal body strength on the effective range of fracturing.** Fig 8A–8D show the simulated results of the phase change fracture of liquid CO$_2$ in the coal seam with tensile strengths of 0.2, 0.5, 0.8 and 1.1 MPa, respectively. The effective radius of fracture was measured to be approximately 357, 354, 352 and 351 cm, and the fracture degree of the coal body was approximately 6.35, 6.34, 6.30 and 6.29%, respectively.

The simulation results show that the effective radius of cracking and the degree of rupture changed slightly with increasing tensile strength. The reason was that when the tensile stress generated by the airburst load was greater than the tensile strength of the coal body, the coal body could be stretched and ruptured, forming cracks; however, the compressive stress generated by the in situ stress suppressed crack generation and expansion, the tensile strength of the coal body was 1–2 orders of magnitude lower than the in situ stress, and crack generation and expansion were mainly to overcome the ground stress, so the tensile strength of the coal body had little effect on its airburst fracture radius and rupture degree.

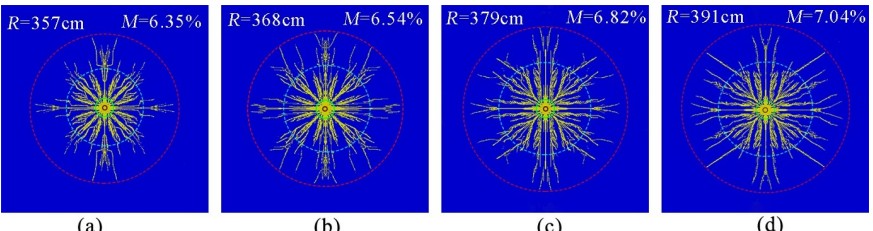

R=357cm M=6.35%  R=368cm M=6.54%  R=379cm M=6.82%  R=391cm M=7.04%

(a)　　　　　(b)　　　　　(c)　　　　　(d)

**Fig 6. Crack distribution under different gas pressures.** (a)Gas pressure 2 MPa.(b)Gas pressure 4 MPa.(c)Gas pressure 6 MPa.(d)Gas pressure 8 MPa.

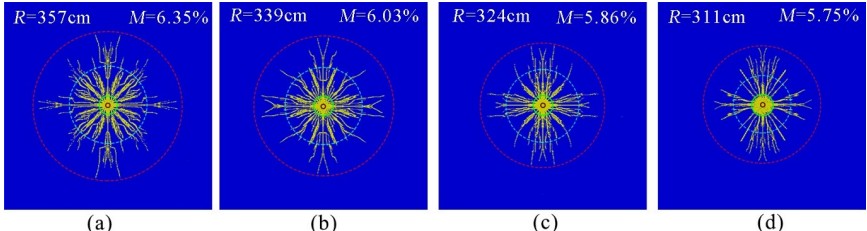

**Fig 7. Crack distribution at different elastic moduli.** (a)Elastic modulus 6.5 MPa.(b)Elastic modulus 4.5 MPa.(c) Elastic modulus 2.5 MPa (d)Elastic modulus 0.5 MPa.

Fig 9A–9D show the simulation results of the phase change fracture of liquid $CO_2$ in the coal seam with compressive strengths of 20, 16, 12 and 8 MPa, respectively. The effective radius of fracture was measured to be approximately 357, 367, 381, and 396cm, and the coal body fracture degree was 6.35, 6.56, 6.81, and 7.09%, respectively.

The simulation results show that the effective radius of cracking and the degree of rupture increased as the compressive strength decreased. The reason was that the greater the compressive strength of coal was, the higher the degree of integrity of the coal rock, the more airburst energy required for crushing, the more energy consumed in the crushing zone, and the same burst energy effect, which would lead to a substantial reduction in the burst energy obtained by the crack dense zone and the crack extension zone, shortening the action time of the stress wave, which was not conducive to the development and extension of the initial crack. In addition, according to the literature [23], the fracture toughness of the coal body increases with the compressive strength, which inhibited the further extension of cracks at a later stage. Therefore, for coal seams with higher compressive strength, the blast energy should be increased appropriately to obtain more effective penetration enhancement.

**3.2.5 Analysis of the effect of pore size on the effective range of fracturing.** Fig 10A–10D show the simulation results of the phase change fracturing of liquid $CO_2$ in a coal seam with hole diameters of 133, 113, 94 and 75 mm, respectively. The effective radius of fracture was measured to be approximately 357, 344, 331, and 322cm; the coal body fracture degrees were approximately 6.35, 6.08, 5.63, and 5.36%, respectively.

The simulation results show that the effective radius of fracture and the degree of rupture decreased with the reduction in pore size. The reason was that as the fracturing pore size increased, the airburst transferred more energy to the surrounding air, which reduced the impact pressure on the hole wall [17] and reduced the energy consumed in the coal body around the borehole to undergo excessive fragmentation and plastic deformation, increasing the use of burst energy in the fracture-intensive and extended areas. In addition, the wave impedance value of air was much smaller than that of rock, and the shock wave of the

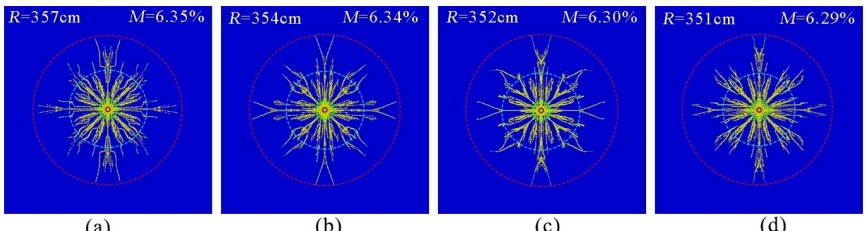

**Fig 8. Crack distribution at different tensile strengths.** (a)Tensile strength 0.2 MPa.(b)Tensile strength 0.5 MPa.(c) Tensile strength 0.8 MPa (d)Tensile strength 1.1 MPa.

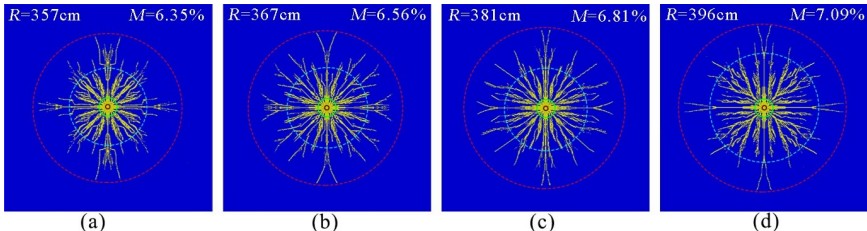

**Fig 9. Crack distribution at different compressive strengths.** (a)Compressive strength 20 MPa.(b)Compressive strength 16 MPa.(c)Compressive strength 12 MPa.(d)Compressive strength 8 MPa.

explosion was buffered as it propagated in the air, which made the action time of the explosion shock wave and stress wave relatively weak, increasing the quasi-static pressure action time of the burst gas on the cracked area, which gave the cracks more time to expand and played an important role in improving the blasting effect. In actual engineering, as the diameter of the drill hole increases, the drilling speed of the drilling machine will decrease. Therefore, to form a better quality precrack and fully consider the work efficiency, the size of the fracturing aperture should be reasonably chosen according to specific conditions (charge quantity, rock properties, etc.) during the construction process.

**3.2.6 Analysis of the influence of the peak value of explosion relief pressure on the effective range of fracturing.** Fig 11A–11D show the simulation results of coal seam liquid $CO_2$ phase change fracturing with the peak values of the explosion relief pressure taken as 180, 210, 240, and 270 MPa, respectively. The effective radius of fracturing was measured to be approximately 357, 371, 386, and 404 cm, and the fracture degree of the coal body was 6.35%, 6.76%, 7.13%, and 7.48%, respectively.

The simulation results show that the effective radius of fracture and the degree of rupture increased with increasing peak pressure of the release blast. The reason was that as the peak pressure of the release increased, the peak value of the dynamic stress intensity factor at the moving tip of the crack increased after the explosion, which caused the peak crack expansion rate to increase and favored an increase in crack length, which was conducive to an increase in crack length. In addition, as the peak pressure of the blast release increased, the dynamic effect of the blast stress wave and the quasi-static effect of the explosive-generated gas were continuously enhanced, and the area of the crush zone and the dense zone of cracks around the gun hole were increased, but the pressure peak of the liquid $CO_2$ blast release was 2–3 orders of magnitude smaller than the pressure peak of the explosive release [15, 17], so it did not cause a large area of the crush zone. Therefore, to obtain a more ideal gas explosion fracturing effect, actual engineering should try to choose a cracker with a larger gas explosion peak pressure.

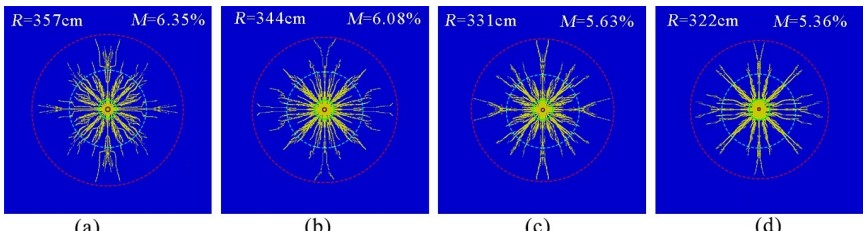

**Fig 10. Crack distribution at different pore sizes.** (a)Pore size 133 mm.(b)Pore size 113 mm.(c)Pore size 94 mm.(d) Pore size 75 mm.

**Fig 11. Crack distribution at different peak relief pressure.** (a)Pressure peak 180 MPa.(b)Pressure peak 210 MPa.(c) Pressure peak 240 MPa (d)Pressure peak 270 MPa.

## 4. Gray correlation analysis of factors affecting the cracking effect of liquid CO$_2$ phase change

The gray correlation analysis method [29] is used to calculate the gray correlation between the data series of system characteristic variables and the data series of related factor variables to derive the order of the influencing factors and finally determine the main influencing factors. Based on the simulation results, this paper took ground stress, gas pressure, dynamic elastic modulus, compressive strength, pore size, and peak vent pressure as the relevant variables $x_1$, $x_2, x_3, x_4, x_5$, and $x_6$. The effective fracture radius and fracture degree, which reflected the superiority of the blasting effect, were used as the systematic characteristic variables $x_\alpha$ and $x_\beta$. Since the tensile strength had little effect on the cracking effect, it was not considered a relevant factor variable in the gray correlation analysis.

The specific calculation steps of the gray correlation analysis of the influencing factors of the phase change cracking effect of liquid CO$_2$ [29] are as follows:

(1) Determining the analysis sequence

The comparison series $X_1, X_2, X_3, X_4, X_5$, and $X_6$ that affect system behavior can be expressed as:

$$\begin{bmatrix} X_1 \\ X_2 \\ X_3 \\ X_4 \\ X_5 \\ X_6 \end{bmatrix}^{\mathrm{T}} = \begin{bmatrix} 15 & 18 & 21 & 24 & 15 & 15 & 15 & 15 & 15 & 15 & 15 & 15 & 15 & 15 & 15 & 15 & 15 & 15 & 15 \\ 2 & 2 & 2 & 2 & 4 & 6 & 8 & 2 & 2 & 2 & 2 & 2 & 2 & 2 & 2 & 2 & 2 & 2 & 2 \\ 6.5 & 6.5 & 6.5 & 6.5 & 6.5 & 6.5 & 6.5 & 4.5 & 2.5 & 0.5 & 6.5 & 6.5 & 6.5 & 6.5 & 6.5 & 6.5 & 6.5 & 6.5 & 6.5 \\ 20 & 20 & 20 & 20 & 20 & 20 & 20 & 20 & 20 & 20 & 16 & 12 & 8 & 20 & 20 & 20 & 20 & 20 & 20 \\ 133 & 133 & 133 & 133 & 133 & 133 & 133 & 133 & 133 & 133 & 133 & 133 & 133 & 113 & 94 & 75 & 133 & 133 & 133 \\ 180 & 180 & 180 & 180 & 180 & 180 & 180 & 180 & 180 & 180 & 180 & 180 & 180 & 180 & 180 & 180 & 210 & 240 & 270 \end{bmatrix}^{\mathrm{T}}$$

The reference sequences $X_\alpha$ and $X_\beta$ that reflect the behavior characteristics of the system can be expressed as:

$$\begin{bmatrix} X_\alpha \\ X_\beta \end{bmatrix}^{\mathrm{T}} = \begin{bmatrix} 357 & 335 & 317 & 308 & 368 & 379 & 391 & 339 & 324 & \hookleftarrow \\ 311 & 367 & 381 & 396 & 344 & 331 & 322 & 371 & 383 & 394 \\ 6.35 & 5.96 & 5.63 & 5.22 & 5.54 & 6.82 & 7.04 & 6.03 & 5.86 & \hookleftarrow \\ 8.75 & 6.56 & 6.81 & 7.09 & 6.08 & 5.61 & 5.36 & 5.76 & 7.13 & 7.48 \end{bmatrix}^{\mathrm{T}}_{2 \times 19}$$

(2) Dimensionless data processing

Since the data in each factor column in the system may not be directly comparable because of different magnitudes, to ensure that all data can participate equally in the calculation, this paper performed the dimensionless processing of data by the method of homogenization, and the calculation formula can be expressed as follows [29].

$$x'_i(k) = \frac{x_i(k)}{\frac{1}{m}\sum\limits_{k}^{m} x_i(k)} \tag{22}$$

where $i$ is the i-th element of the test sample, $k$ is the k-th indicator of the test element, and $m$ is the number of data points in a single sequence.

This paper is based on the original data set of factors influencing the liquid CO$_2$ blasting and blasting effect. Substituting the corresponding values, the dimensionless data sequence forms the following matrix.

The comparison sequence $X'_1$, $X'_2$, $X'_3$, $X'_4$, $X'_5$, and $X'_6$ after dimensionless processing can be expressed as follows.

$$\begin{bmatrix} X'_1\ X'_2\ X'_3\ X'_4\ X'_5\ X'_6 \end{bmatrix} = \begin{bmatrix}
0.9406 & 0.7600 & 1.1076 & 1.0674 & 1.0485 & 0.9500 \\
1.1287 & 0.7600 & 1.1076 & 1.0674 & 1.0485 & 0.9500 \\
1.3168 & 0.7600 & 1.1076 & 1.0674 & 1.0485 & 0.9500 \\
1.5050 & 0.7600 & 1.1076 & 1.0674 & 1.0485 & 0.9500 \\
0.9406 & 1.5200 & 1.1076 & 1.0674 & 1.0485 & 0.9500 \\
0.9406 & 2.2800 & 1.1076 & 1.0674 & 1.0485 & 0.9500 \\
0.9406 & 3.0400 & 1.1076 & 1.0674 & 1.0485 & 0.9500 \\
0.9406 & 0.7600 & 0.7668 & 1.0674 & 1.0485 & 0.9500 \\
0.9406 & 0.7600 & 0.4260 & 1.0674 & 1.0485 & 0.9500 \\
0.9406 & 0.7600 & 0.0852 & 1.0674 & 1.0485 & 0.9500 \\
0.9406 & 0.7600 & 1.1076 & 0.8539 & 1.0485 & 0.9500 \\
0.9406 & 0.7600 & 1.1076 & 0.6404 & 1.0485 & 0.9500 \\
0.9406 & 0.7600 & 1.1076 & 0.4270 & 1.0485 & 0.9500 \\
0.9406 & 0.7600 & 1.1076 & 1.0674 & 0.8909 & 0.9500 \\
0.9406 & 0.7600 & 1.1076 & 1.0674 & 0.7411 & 0.9500 \\
0.9406 & 0.7600 & 1.1076 & 1.0674 & 0.5913 & 0.9500 \\
0.9406 & 0.7600 & 1.1076 & 1.0674 & 1.0485 & 1.1083 \\
0.9406 & 0.7600 & 1.1076 & 1.0674 & 1.0485 & 1.2667 \\
0.9406 & 0.7600 & 1.1076 & 1.0674 & 1.0485 & 1.4250
\end{bmatrix}$$

The reference sequences $X'_\alpha$ and $X'_\beta$ after dimensionless processing can be represented as

follows.

$$\begin{bmatrix} X'_{\alpha} \\ X'_{\beta} \end{bmatrix}^{\text{T}} = \begin{bmatrix} 1.0097 & 0.9475 & 0.8965 & 0.8711 & 1.0408 & 1.0719 & 1.1058 & 0.9588 & 0.9163 & \hookleftarrow \\ 0.8796 & 1.0380 & 1.0776 & 1.1200 & 0.9729 & 0.9361 & 0.9107 & 1.0493 & 1.0832 & 1.1143 \\ 1.0047 & 0.9430 & 0.8908 & 0.8259 & 1.0348 & 1.0791 & 1.1139 & 0.9541 & 0.9272 & \hookleftarrow \\ 0.9098 & 1.0380 & 1.0775 & 1.1218 & 0.9620 & 0.8877 & 0.8481 & 1.0696 & 1.1282 & 1.1835 \end{bmatrix}_{2 \times 19}^{\text{T}}$$

By dimensionless processing, the formal unification of the original data series was accomplished, and the initial maximum absolute difference of the data was weakened to avoid the interference and influence of the extreme data on the calculation process.

(3) Calculation of the correlation coefficient

The correlation coefficient, the relative difference of the data factor series, is calculated by a process that relies on the absolute value data of all differences in the single series of each influence factor, as well as the two-level minimum and two-level maximum differences of the data series of all influence factors, calculated as follows [29].

$$\zeta_i(k) = \frac{\Delta \min + \rho \Delta \min}{\left| x'(k) - x'_i(k) \right| + \rho \Delta \min} \quad (i = 1, 2, 3, 4, 5, 6) \tag{23}$$

where $\xi$ is the correlation coefficient, $\rho$ is the resolution factor, there are no significant gaps in the data series in this paper, and there is no rank difference in data acquisition; therefore, $\rho$ is taken to be 0.5. $\left| x'(k) - x'_i(k) \right|$ is the absolute difference between the dimensionless processed reference series and the comparison series at the $k$-th indicator of the $i$-th element, and $\Delta$min and $\Delta$max are the minimum and maximum differences between the two levels, respectively.

$\Delta$min and $\Delta$max can be expressed as:

$$\Delta \min = \min_i \min_k \left| x'(k) - x'_i(k) \right| \quad (i = 1, 2, 3, 4, 5, 6) \tag{24}$$

$$\Delta \max = \max_i \max_k \left| x'(k) - x'_i(k) \right| \quad (i = 1, 2, 3, 4, 5, 6) \tag{25}$$

The $\Delta$min (R) and $\Delta$max (R) of the fracture-causing effective radius variable and the associated factor variables are obtained from the data in the text.

$$\Delta \min(R) = \min_i \min_k \left| x'_{\alpha}(k) - x'_i(k) \right| = 0.00072$$

$$\Delta \max(R) = \max_i \max_k \left| x'_{\alpha}(k) - x'_i(k) \right| = 1.9342$$

The $\Delta$min (M) and $\Delta$max (M) of the cracking and factor-related variables can be expressed as follows.

$$\Delta \min(M) = \min_i \min_k \left| x'_{\beta}(k) - x'_i(k) \right| = 0.0022$$

$$\Delta \max(M) = \max_i \max_k \left| x'_{\beta}(k) - x'_i(k) \right| = 1.9261$$

The above calculation results are brought into Eq (23) to obtain the correlation coefficient $\xi \alpha i$

between each influencing factor and the effective radius of fracture causing.

$$
[\zeta_{\alpha1}\ \zeta_{\alpha2}\ \zeta_{\alpha3}\ \zeta_{\alpha4}\ \zeta_{\alpha5}\ \zeta_{\alpha6}] =
\begin{bmatrix}
0.9340 & 0.7954 & 0.9087 & 0.9444 & 0.9621 & 0.9426 \\
0.8428 & 0.8383 & 0.8586 & 0.8903 & 0.9060 & 0.9981 \\
0.6976 & 0.8769 & 0.8215 & 0.8505 & 0.8648 & 0.9483 \\
0.6045 & 0.8976 & 0.8041 & 0.8319 & 0.8456 & 0.9253 \\
0.9068 & 0.6692 & 0.9361 & 0.9739 & 0.9928 & 0.9149 \\
0.8811 & 0.4449 & 0.9651 & 0.9961 & 0.9772 & 0.8887 \\
0.8547 & 0.3336 & 0.9989 & 0.9625 & 0.9448 & 0.8619 \\
0.9823 & 0.8301 & 0.8350 & 0.8997 & 0.9157 & 0.9918 \\
0.9763 & 0.8615 & 0.6641 & 0.8655 & 0.8804 & 0.9671 \\
0.9414 & 0.8906 & 0.5494 & 0.8380 & 0.8519 & 0.9328 \\
0.9092 & 0.7773 & 0.9335 & 0.8408 & 0.9899 & 0.9173 \\
0.8766 & 0.7534 & 0.9706 & 0.6892 & 0.9716 & 0.8841 \\
0.8442 & 0.7293 & 0.9881 & 0.5830 & 0.9319 & 0.8511 \\
0.9684 & 0.8202 & 0.8784 & 0.9117 & 0.9225 & 0.9776 \\
0.9962 & 0.8466 & 0.8500 & 0.8811 & 0.8328 & 0.9866 \\
0.9707 & 0.8658 & 0.8314 & 0.8612 & 0.7523 & 0.9617 \\
0.8996 & 0.7703 & 0.9438 & 0.9823 & 1.0000 & 0.9431 \\
0.8721 & 0.7501 & 0.9761 & 0.9847 & 0.9661 & 0.8412 \\
0.8484 & 0.7324 & 0.9939 & 0.9545 & 0.9370 & 0.7574
\end{bmatrix}
$$

The matrix form of the correlation coefficient $\xi_{\beta i}$ of each influencing factor and the degree of cracking are as follows:

$$
[\zeta_{\beta1}\ \zeta_{\beta2}\ \zeta_{\beta3}\ \zeta_{\beta4}\ \zeta_{\beta5}\ \zeta_{\beta6}] =
\begin{bmatrix}
0.9397 & 0.7992 & 0.9056 & 0.9411 & 0.9587 & 0.9484 \\
0.8403 & 0.8422 & 0.8560 & 0.8876 & 0.9033 & 0.9951 \\
0.6949 & 0.8824 & 0.8181 & 0.8470 & 0.8612 & 0.9443 \\
0.5878 & 0.9381 & 0.7755 & 0.8014 & 0.8141 & 0.8879 \\
0.9130 & 0.6665 & 0.9318 & 0.9695 & 0.9882 & 0.9212 \\
0.8763 & 0.4461 & 0.9735 & 0.9903 & 0.9715 & 0.8838 \\
0.8494 & 0.3341 & 0.9958 & 0.9561 & 0.9386 & 0.8565 \\
0.9884 & 0.8342 & 0.8391 & 0.8968 & 0.9128 & 0.9980 \\
0.9886 & 0.8540 & 0.6592 & 0.8749 & 0.8901 & 0.9791 \\
0.9712 & 0.8674 & 0.5400 & 0.8613 & 0.8761 & 0.9621 \\
0.9102 & 0.7778 & 0.9347 & 0.8415 & 0.9914 & 0.9184 \\
0.8775 & 0.7538 & 0.9719 & 0.6894 & 0.9730 & 0.8851 \\
0.8435 & 0.7286 & 0.9877 & 0.5822 & 0.9314 & 0.8505 \\
0.9805 & 0.8285 & 0.8707 & 0.9034 & 0.9333 & 0.9899 \\
0.9501 & 0.8850 & 0.8159 & 0.8446 & 0.8699 & 0.9414 \\
0.9145 & 0.9183 & 0.7895 & 0.8164 & 0.7913 & 0.9064 \\
0.8839 & 0.7584 & 0.9642 & 1.0000 & 0.9808 & 0.9636 \\
0.8389 & 0.7251 & 0.9814 & 0.9428 & 0.9258 & 0.8763 \\
0.8004 & 0.6961 & 0.9290 & 0.8944 & 0.8791 & 0.8014
\end{bmatrix}
$$

(4) Calculation of the correlation

The gray correlation is used to make the data tend to be in equilibrium, which is a description of the data from a system theory perspective seeking an intrinsic correlation of things. The correlation of each influencing factor of liquid CO$_2$ phase change fracturing for the fracturing effect element was calculated as [29]:

$$\gamma_i = \frac{1}{m} \sum_{k=1}^{m} \zeta_i(k) \tag{26}$$

where $\gamma$ is the correlation degree.

The resulting gray correlation coefficients were substituted in Eq (26), and the gray correlations of the phase change fracturing of liquid CO$_2$ in coal rocks were obtained, as shown in Table 4.

The Table 4 gray correlation results show the gray correlation cumulative ranking of all factors: the peak pressure of the release blast ($\gamma$ = 1.8421) was greater than the hole diameter ($\gamma$ = 1.8333), the hole diameter was greater than the ground stress ($\gamma$ = 1.7609), the ground stress was greater than the compressive strength ($\gamma$ = 1.7517), the compressive strength was greater than the modulus of elasticity ($\gamma$ = 1.7498), the modulus of elasticity was greater than the gas pressure ($\gamma$ = 1.5273). Whether from the gray correlation of individual indicators or the gray correlation of the integrated indicators, the peak pressure of the release of the effective radius of fracture and fracture degree of blasting had the greatest impact, and it was obviously the main control factor of the six factors affecting the fracturing effect. Therefore, when the blasting effect is optimized, the peak value of the blast relief pressure of the cracker should be adjusted first. At the top of the cumulative ranking of the gray correlation degree was the peak vent pressure and the size of the fracturing aperture, which showed that the blasting parameters had a greater influence than the coal physical parameters on the blasting effect. This conclusion is consistent with the conclusion in the literature [30]. Among the effects of the physical parameters of the coal body on the blasting effect, the gray correlation index of ground stress had the greatest influence, which indicated that the rupture and fracture development of coal rock under blasting loading was dominated by overcoming ground stress. The effect of gas pressure on the fracturing effect was smaller than that of other factors, but its influence cannot be ignored when blasting in high-gas coal seams.

In summary, coal physical parameters and blasting parameters have different degrees of influence on the blasting effect, but in different blasting projects, coal physical parameters can only be surveyed but not changed. Therefore, when liquid CO$_2$ is blasted and designed, blasting parameters must be correctly selected on the basis of a detailed survey of coal physical parameters to determine reasonable blasthole spacing and location and thus to obtain the best blasting effect.

**Table 4. Gray correlation of phase change fracturing of liquid CO$_2$ in coal rocks.**

| Reference variable | Compared variables | | | | | |
|---|---|---|---|---|---|---|
| | $x_1$ | $x_2$ | $x_3$ | $x_4$ | $x_5$ | $x_6$ |
| $x_\alpha$ | 0.8846 | 0.7623 | 0.8793 | 0.8811 | 0.9182 | 0.9206 |
| $x_\beta$ | 0.8763 | 0.7650 | 0.8705 | 0.8706 | 0.9153 | 0.9215 |
| summation | 1.7609 | 1.5273 | 1.7498 | 1.7517 | 1.8333 | 1.8421 |

## 5. Engineering verification of the phase change fracturing effect of liquid CO$_2$

### 5.1 Geological conditions of the coal seam in the test area

This paper takes the No. 15 coal seam 203 working face of the Mabao coal mine in Shanxi Province, China, as the test working face. The ground elevation of this working face is +915~ +825 m, the thickness of the coal seam is 4.8~6.2 m, averaging 5.5 m, the dip angle of the coal seam is 0~4˚, the original gas pressure of the coal seam is large, reaching a maximum of 4.6 MPa, the ground stress is 16 MPa, the elastic modulus of the coal body is 3.2 GPa, the compressive strength is 14 MPa, the tensile strength is 0.45 MPa, the maximum raw coal gas content is 10.5 m$^3$/t, the absolute gas emission from the working face is 18 m$^3$/min, and the permeability coefficient of the coal seam is 0.32 m$^2$/(MPa$^2$ d), which is a high-gas recoverable coal seam.

### 5.2 Experimental design

According to the geological conditions of the test face, this paper investigated the effect of phase change fracturing of liquid CO$_2$ in coal seams by changing the fracture hole diameter and the peak pressure of the fracture release, and set up five blasting test groups at 500, 550, 600, 650 and 700 m from the coal seam cuttings, using the same layout, with one fracture hole and eight observation holes in each group, as shown in Fig 12. The parameters of the cloth holes are shown in Table 5. In the process of drilling construction, 8 observation holes in the experimental group needed to be constructed first, and the fracturing holes were constructed last. After constructing the observation holes, SF$_6$ transmitters were sent into the holes, the holes were sealed by the method of "two blocking and one injection," and pressure gauges and gas concentration recorders were installed after the holes were sealed, as shown in Fig 13. After the liquid CO$_2$ phase change blasting was completed, the hole was first sealed, then the tracer gas SF$_6$ was injected into the fracture hole, and recording began.

### 5.3 Analysis of test results

**5.3.1 Analysis of effective radius test results of coal seam fracturing.** In the on-site monitoring process, the data obtained are the change in the voltage value transmitted by the transmitter, which needs to be converted into a gas concentration value [31]. The conversion formula is as follows:

$$C = \kappa * U^2 \tag{27}$$

where $C$ is the gas concentration value in cm$^3$/m$^3$; $\kappa$ is a coefficient with a value of 3.1; and $U$ is the signal voltage transmitted by the transmitter in V.

Through data processing, the continuous change process of the SF$_6$ gas concentration value in the observation hole could be obtained within 30 days. According to the method provided in the literature [31] of using tracer gas SF$_6$ to measure the permeability coefficient of the coal

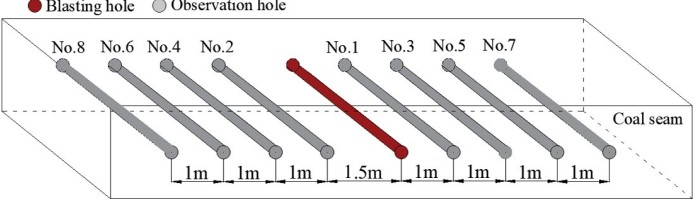

**Fig 12. Schematic diagram of the drilling arrangement of the test working face.**

**Table 5. Test hole layout parameters.**

| Drill hole number | aperture/mm | slope/(°) | hole depth/m | sealing length/m | Peak burst pressure/MPa |
|---|---|---|---|---|---|
| Group 1 blasting holes | 94 | +5 | 60 | 16 | 180 |
| Group 2 blasting holes | 94 | +5 | 60 | 16 | 210 |
| Group 3 blasting holes | 94 | +5 | 60 | 16 | 240 |
| Group 4 blasting holes | 113 | +5 | 60 | 16 | 240 |
| Group 5 blasting holes | 133 | +5 | 60 | 16 | 240 |
| All observation holes | 113 | +5 | 60 | 16 | —— |

seam, the data of the permeability coefficient $\lambda$ of 40 observation holes in the 5 groups of blasting test groups could be obtained by calculation, as shown in Table 6. The distance between each group of observation holes and the fracture hole was used as the dependent variable, and the coefficient of coal seam permeability was used as an independent variable. The rate of change of coal seam permeability at different distances from the fracture hole was obtained by first-order differentiation using mathematical software, as shown in Fig 14. The rate of change of permeability near 0 was used as the effective radius of cracking.

As seen from Table 6, after liquid CO$_2$ blasting, the coal seam permeability increased by 6.4~22.1 times within 2 m, indicating that this area was a crack dense area and that the coal seam obtained a better effect of increasing permeability. The improvement rate of coal seam permeability gradually decreased outside 2.0 m until it gradually returned to the original coal seam permeability outside 3.5 m, indicating that this area was the crack extension area. Fig 14 shows that when the fracturing aperture was 94 mm, the peak vent pressure of the fracturing device was 180, 210, and 240 MPa, and the corresponding effective fracturing radii were approximately 3.34, 3.49, and 3.64 m, respectively. When the peak vent pressure was 240 MPa and the fracture aperture was 94, 113, and 133 mm, the corresponding fracturing effective radii were approximately 3.64, 3.77, and 3.89 m, respectively. The effect of field liquid CO$_2$ blasting showed that the effective radius of fracture increased with the peak pressure of release and the fracture hole diameter, which was consistent with the conclusions obtained from the simulation results of this paper.

**5.3.2 Analysis of scanning electron microscopy micromorphological feature test results.** Scanning electron microscopy (SEM) was used to analyze the microscopic morphological characteristics of the original coal samples and five sets of liquid CO$_2$ phase change fractured coal samples from industrial tests. Fig 15 shows the results of the test at 3000x magnification.

The surface micromorphological scan results of the original coal sample in Fig 15A, show that the surface of the original coal sample was relatively smooth and contained a few smaller pores without obvious rupture. Fig 15B shows the surface microscopic morphology of the

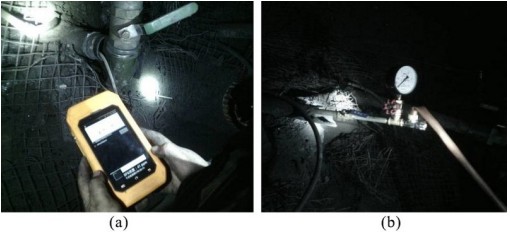

(a)                                (b)

**Fig 13. *In situ* determination of tracer gas SF$_6$.** (a) Gas concentration recorder. (b) Pressure gauge.

**Table 6. Permeability coefficients of observation holes at different distances from fracture holes.**

| Distance from the fracture-causing hole/m | | 1 | 1.5 | 2.0 | 2.5 | 3.0 | 3.5 | 4.0 | 4.5 |
|---|---|---|---|---|---|---|---|---|---|
| penetrability λ/m$^{2.}$ (MPa$^2$ ·d)$^{-1}$ | group 1 | 4.55 | 2.15 | 1.44 | 0.56 | 0.35 | 0.32 | 0.32 | 0.32 |
| | group 2 | 5.28 | 3.76 | 1.95 | 0.84 | 0.36 | 0.34 | 0.32 | 0.32 |
| | group 3 | 6.08 | 3.89 | 2.06 | 1.37 | 0.60 | 0.33 | 0.32 | 0.32 |
| | group 4 | 6.59 | 4.02 | 2.23 | 1.42 | 0.57 | 0.35 | 0.32 | 0.32 |
| | group 5 | 7.06 | 4.73 | 2.72 | 1.48 | 0.82 | 0.36 | 0.32 | 0.32 |

group 1 test coal sample, and it can be seen from the figure that the surface of the coal body cracked by the phase change of liquid CO$_2$ had a blocky distribution morphology, and a crack with a length of approximately 14 μm was present in the coal rock at the bottom right of the image although it was relatively flat, and the rupture degree of the coal body surface had obviously increased compared with the original coal sample. As shown in Fig 15C, the surface of the coal samples tested in Group 2 was uneven, with more pores and increased fracture compared to the coal samples tested in Group 1, and there were no areas of large flatness. Fig 15D shows the surface micromorphology of the test coal sample of group 3. The surface damage below the coal sample was more severe, with more pores, larger pore diameters, and smaller blast sizes, and the maximum diameter of the broken pieces was approximately 10 μm. Fig 15E shows the surface micromorphology of the coal sample of the group 4 test. Compared with the coal sample of the group 3 test, the whole surface of the coal sample was severely damaged, the number of pores increased, the pore diameters increased, the blast block decreased, and the maximum diameter of the broken block was approximately 7 μm. As shown in Fig 15F, the surface pore distribution of the coal samples tested in group 5 was dense compared with that of the coal samples tested in group 4, the blast block size was substantially lower, and the maximum diameter of the broken pieces was approximately 3 μm, indicating that the structural rupture of the coal samples was the most severe under the air blast load.

Based on the above analysis, the SEM images of liquid CO$_2$ phase change fractured coal samples in Fig 15B–15D show that as the peak fracture pressure of the fracturing device increased, the fracture degree of the coal seam also increased. The SEM images in Fig 15D–15F show that the fracture degree of the coal seam increased with the fracture aperture. The SEM image test results of the coal samples are consistent with the conclusions obtained from the simulation results in this paper.

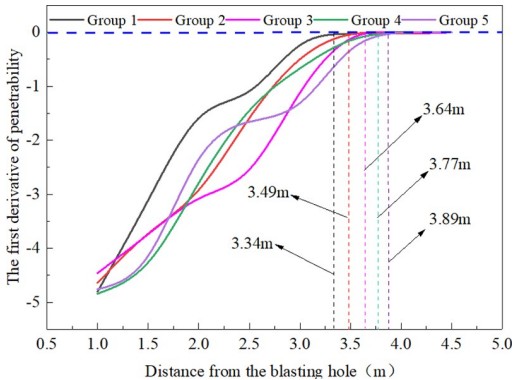

**Fig 14. First derivative curve of the permeability of the coal seam.**

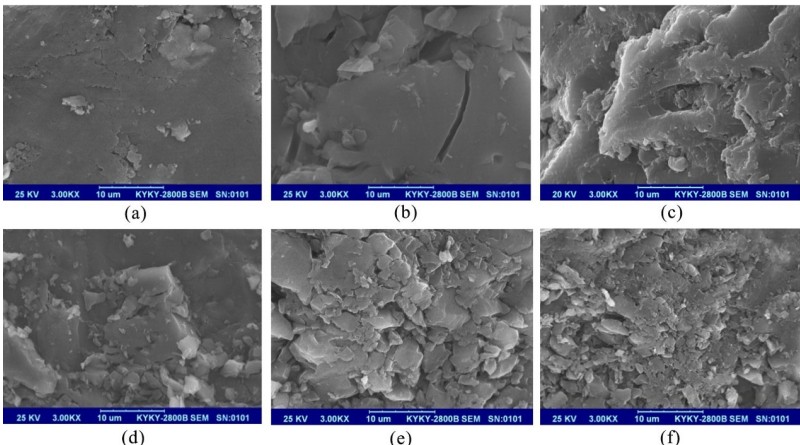

**Fig 15. SEM test results of coal samples with different blasting parameters.** (a)Original coal sample.(b)Group 1 coal sample.(c)Group 2 coal sample(d)Group 3 coal sample.(e)Group 4 coal sample.(f)Group 5 coal sample.

## 5.3 Comparison of simulation results with experimental results

Combined with the geological conditions of the coal seam in the test area, the blasting parameters of five test groups were input into the numerical model developed in this paper to test the effective radius of fracture, and the test results are shown in Table 7.

Table 7 shows that the simulation results of liquid CO$_2$ phase change fracturing in the coal seam were smaller than the industrial test results. Because of the complexity of the coal body itself, different coal bodies had different degrees of primary fractures, and their internal crevices and joints were intricate and complex, which played a certain role in inducing high-pressure gas to produce fractures in the coal body. Moreover, the roadway excavation and drilling construction also had an impact on the nearby coal body, so the actual media expansion range in the field was large, while the results of the numerical simulation were small, and there was a certain error, but the relative error was less than 10%, which could meet the needs of engineering applications. Overall, the numerical test results basically match the field test results, indicating that the numerical model of coal rock blasting established in this paper is reasonable and reliable.

## 6. Conclusions

This thesis focuses on the current situation of difficult gas extraction in low-permeability and high-gas coal seams under complex geological conditions, and it takes the factors influencing the fracturing effect of liquid CO$_2$ phase change in coal seams as the research object, using a combination of theoretical analysis, numerical simulation and field industrial comparison tests, conducting a study on the mechanical mechanism of coal rock crack extension under the action of liquid CO$_2$ phase change blasting load and simulating and analyzing the effect of physical parameters of coal seam and blasting parameters on the effect of liquid CO$_2$ phase

**Table 7. Comparison of experimental results and simulation results.**

| Test number | | 1 | 2 | 3 | 4 | 5 |
|---|---|---|---|---|---|---|
| Effective radius of cracking/m | Industrial trials | 3.34 | 3.49 | 3.64 | 3.77 | 3.89 |
| | Numerical simulation | 3.19 | 3.26 | 3.51 | 3.53 | 3.71 |
| Relative Error/% | | 4.49 | 6.59 | 3.57 | 6.37 | 4.63 |

change cracking. The degree of influence of each factor was also analyzed by using gray correlation theory and based on the above study, an underground industrial comparison experiment was implemented in the Mabao coal mine, Shanxi, China. The main conclusions obtained from this paper are as follows.

1. The mechanism of liquid CO$_2$ phase change fracture was analyzed, the calculation method of liquid CO$_2$ phase change fracture TNT equivalent was determined, the liquid CO$_2$ phase change airburst loading process was elaborated, and the mechanism of coal rock crack generation and extension under airburst loading was studied, which provided the theoretical basis for the numerical simulation of liquid CO$_2$ phase change airburst.

2. By using simulation software, the effect of phase change fracturing of liquid CO$_2$ was positively correlated with gas pressure, modulus of elasticity, fracture hole diameter and peak pressure of explosion release and negatively correlated with ground stress and compressive strength. The tensile strength had little effect on the cracking effect, which provided theoretical guidance for selecting blasting parameters and optimizing the spacing of holes in engineering construction.

3. According to gray correlation analysis, liquid CO$_2$ phase change blasting parameters had a greater effect than coal seam physical parameters on the fracturing effect. The degree of influence of each influencing factor on the phase change fracturing effect of liquid CO$_2$ is ranked as follows: the peak pressure of the release blast was greater than the hole diameter, the hole diameter was greater than the in situ stress, the ground stress was greater than the compressive strength, the compressive strength was greater than the modulus of elasticity, and the modulus of elasticity was greater than the gas pressure. During construction, blasting parameters should be designed based on reliable physical parameters of the coal seam to obtain more effective penetration enhancement.

4. Phase change blasting tests of liquid CO$_2$ in coal seams under different blasting parameters showed that when the peak fracture release pressure was 180, 210, and 240 MPa, the corresponding fracture effective radii were 3.34, 3.49, and 3.64 m, respectively; when the fracture hole diameters were 94, 113 and 133 mm, the corresponding fracture effective radii were approximately 3.64, 3.77 and 3.89 m, respectively. The effective fracture radius of coal rock was positively correlated with the peak relief pressure and fracture hole diameter. SEM test results of coal samples with different blasting parameters showed that the degree of coal rock rupture was positively correlated with the peak blast pressure and fracture hole diameter. The error between the engineering test results of the effective radius of fracture and the simulation results was less than 10%, which proved that the numerical model of coal rock blasting established in this paper was reasonable and reliable.

## Author Contributions

**Conceptualization:** Jinzhang Jia, Dongming Wang, Bin Li.

**Data curation:** Jinzhang Jia, Bin Li, Xiuyuan Tian.

**Formal analysis:** Xiuyuan Tian.

**Funding acquisition:** Jinzhang Jia.

**Software:** Dongming Wang, Bin Li.

**Validation:** Xiuyuan Tian.

**Visualization:** Jinzhang Jia.

**Writing – original draft:** Jinzhang Jia, Dongming Wang.

**Writing – review & editing:** Bin Li.

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
