## [Decision Letter · Decision Letter 0]

23 Jun 2021

PONE-D-21-16377

Study of the influencing factors of the liquid CO2 phase change fracturing effect in coal seams

PLOS ONE

Dear Dr. Li,

Thank you for submitting your manuscript to PLOS ONE. After careful consideration, we feel that it has merit but does not fully meet PLOS ONE’s publication criteria as it currently stands. Therefore, we invite you to submit a revised version of the manuscript that addresses the points raised during the review process.

Please consider all comments

We look forward to receiving your revised manuscript.

Kind regards,

Ahmed Mancy Mosa, Ph.D.

Academic Editor

PLOS ONE

Journal Requirements:

2. Please amend either the title on the online submission form (via Edit Submission) or the title in the manuscript so that they are identical.

['This research was supported by the National Natural Science Foundation of China (No. 51374121) and

funded by Liaoning Distinguished Professor (551710007007), funded project of the Liaoning Million Talents

project (2019-45-15), and the Natural Science Foundation of Liaoning Province (2019-MS-162).'

'The funders had no role in study design, data collection and analysis, decision to

publish, or preparation of the manuscript.'

Additional Editor Comments (if provided):

Reviewers' comments:

Reviewer's Responses to Questions

**Comments to the Author**

1. Is the manuscript technically sound, and do the data support the conclusions?

Reviewer #1: Yes

Reviewer #2: Yes

Reviewer #3: Yes

Reviewer #4: Yes

Reviewer #5: Yes

2. Has the statistical analysis been performed appropriately and rigorously? 

Reviewer #1: Yes

Reviewer #2: Yes

Reviewer #3: Yes

Reviewer #4: Yes

Reviewer #5: Yes

3. Have the authors made all data underlying the findings in their manuscript fully available?

Reviewer #1: Yes

Reviewer #2: Yes

Reviewer #3: Yes

Reviewer #4: Yes

Reviewer #5: No

4. Is the manuscript presented in an intelligible fashion and written in standard English?

Reviewer #1: Yes

Reviewer #2: Yes

Reviewer #3: Yes

Reviewer #4: Yes

Reviewer #5: Yes

5. Review Comments to the Author

Reviewer #1: In this paper the current situation of low permeability gas extraction from coal seam gas difficult, will the influence factors of liquid CO2 phase change of coal seam crack effect as the research object, first analyzed theoretically, and through the simulation software to simulate the effects of different factors on the CO2 phase change crack damage effect, after using the mathematical algorithm to sort the factors important degree, finally carried out the field contrast test, This study provides a theoretical basis for the practical construction of liquid CO2 phase change cracking and anti-reflection technology in mines, and has some innovation. However, there are still some contents in the article that need to be modified. Suggestions are as follows:

1. In Section 5.1, the coal seam geological parameters should increase the raw coal gas content and the absolute gas emission amount at the working face.

2. Some of the formula sources in the paper need to add references. For example, Formula (10), because these formulas are not original to the author.

3. Can the liquid CO2 phase change induced cracking and anti-reflection technology described in this paper cause gas explosion?

4. “The effective radius of fracture was measured to be approximately 357, 367, 381, and 396, and the coal body fracture degree was 6.35%, 6.56%, 6.81%, and 7.09%, respectively.” “The effective radius of fracture was measured to be approximately 357, 344, 331, and 322.” The parameters in these two sentences are missing units.

5. Due to the length of the article, it is recommended that there should be no less than 30 references.

6. The "R" and "M" on all simulation diagrams in this paper should be italicized because they are variables.

Reviewer #2: Through the application of simulation software, this study analyzed the influence of physical parameters of coal seam on phase change cracking effect of liquid CO2, and verified the reliability of blasting numerical model through comparative engineering tests of different blasting parameters. The research results have a certain innovative significance for the application of liquid CO2 fracturing technology in coal mines.

The study presents the results of original research. Results reported have not been published elsewhere. Experiments, statistics, and other analyses are performed to a high technical standard and are described in sufficient detail. Conclusions are presented in an appropriate fashion and are supported by the data. The article is presented in an intelligible fashion and is written in standard English. The research meets all applicable standards for the ethics of experimentation and research integrity. The article adheres to appropriate reporting guidelines and community standards for data availability.

Reviewer #3: In this paper, based on the mechanical mechanism of coal rock crack propagation under the blasting load of liquid CO2 phase change, the influence of liquid CO2 phase change cracking effect was studied by means of simulation software, mathematical methods and field tests.The influence of thesis writing logic is stronger, system structure is reasonable, the writing more specification, research results have good practical value, and has certain innovative. However, this article still needs some simple changes. Suggestions are:

1. The difference and advantage of liquid carbon dioxide phase change anti-reflection technology compared with conventional charge blasting anti-reflection technology.

2. The "blasting hole" in Fig.13 and the "fissionable hole" in Table 5 describe a thing, so please be consistent and ensure that the whole article is consistent.

3. Will there be secondary pollution after SF6 gas is injected into coal seam?

4. Some formulas in the literature need to be quoted. Variables in the formulas need to be represented in italics and constants need to be represented in upright letters. Please check the full text to ensure accuracy.

Reviewer #4: This is an interesting paper that contains some valuable new data about the study the liquid CO2 phase change fracturing of coal seam. It also analyzes various influencing factors and their contributions. This has certain guiding function to theory and practice.

Still, the major modifications or clarifications should be made.

Detailed comments

1. It will be more appropriate to change the statement of "grey correlation comprehensive index" to "grey correlation cumulative index".

2. Some mathematical symbols in the formula, if the value is fixed, can not use italics, please correct the English punctuation marks.

3. “2.2.5 Mechanism of coal rock crack extension under liquid CO2 phase change airburst loading”- This part is based on the linear elastic fracture mechanics explains the mechanism of crack extension of the coal. However, the coal is actually not a brittle material, it is a quasi brittle materials. There will be a fracture process zone (FPZ) at the crack tip of a quasi-brittle material. Due to its existence, the linear elastic fracture mechanics is no longer applicable. The influence of FPZ on the fracture process is significant in the static process, and has been extensively studied, but the research in the dynamic process is still insufficient. The author can refer to related references for further explanation. This will be helpful for in-depth understanding of coal fracture process. If possible, it is suggested to increase the effect of fracture process zone on the fracturing effect.

4. The coal seam parameter description in Chapter 5 is a little less, so some other parameters should be added appropriately.

Reviewer #5: This paper studies the effects of different physical parameters ( in situ stress, gas pressure, modulus of elasticity and strength of coal ) and blasting parameters ( fracturing pore size and peak pressure of detonation ) on the phase change fracturing effect of liquid CO2. After reading, I think there are some problems in the structure, format and logic of the manuscript. I suggest that it be revised and published. It mainly includes the following aspects:

(1) There are no keywords in the text.

(2) Some paragraphs of the whole article are malformed, without indenting the first line.

(3) In Section 2.1, title format is wrong.

(4) The title of the chart is not centered and the formula label is not aligned.

(5) Section 3 does not specify the specific simulation software. What is the reference basis for building the model?

(6) The picture is not clear.

(7) There are many paragraphs in the conclusion, which are suggested to be explained in sections.

6. PLOS authors have the option to publish the peer review history of their article (what does this mean?). If published, this will include your full peer review and any attached files.

Reviewer #1: No

Reviewer #2: No

Reviewer #3: No

Reviewer #4: No

Reviewer #5: No

---

## [Author Response · Author response to Decision Letter 0]

28 Jun 2021

A separate response letter has been uploaded

---

## [Decision Letter · Decision Letter 1]

8 Jul 2021

Study of the influencing factors of the liquid CO2 phase change fracturing effect in coal seams

PONE-D-21-16377R1

Dear Dr. Li,

We’re pleased to inform you that your manuscript has been judged scientifically suitable for publication and will be formally accepted for publication once it meets all outstanding technical requirements.

Kind regards,

Ahmed Mancy Mosa, Ph.D.

Academic Editor

PLOS ONE

Additional Editor Comments (optional):

Reviewers' comments:

Reviewer's Responses to Questions

**Comments to the Author**

1. If the authors have adequately addressed your comments raised in a previous round of review and you feel that this manuscript is now acceptable for publication, you may indicate that here to bypass the “Comments to the Author” section, enter your conflict of interest statement in the “Confidential to Editor” section, and submit your "Accept" recommendation.

Reviewer #1: All comments have been addressed

Reviewer #3: All comments have been addressed

Reviewer #4: All comments have been addressed

2. Is the manuscript technically sound, and do the data support the conclusions?

Reviewer #1: Yes

Reviewer #3: Yes

Reviewer #4: Yes

3. Has the statistical analysis been performed appropriately and rigorously? 

Reviewer #1: Yes

Reviewer #3: Yes

Reviewer #4: Yes

4. Have the authors made all data underlying the findings in their manuscript fully available?

Reviewer #1: Yes

Reviewer #3: Yes

Reviewer #4: Yes

5. Is the manuscript presented in an intelligible fashion and written in standard English?

Reviewer #1: Yes

Reviewer #3: Yes

Reviewer #4: Yes

6. Review Comments to the Author

Reviewer #1: To study the influence of different factors on the cracking effect of the liquid CO 2 phase transition, the mechanics of

coal rock crack extension based on liquid CO 2 phase change blast loading were studied. Through the application of

simulation software to analyze the influence of coal seam physical parameters （in situ stress, gas pressure, modulus of

elasticity and strength of coal） and blasting parameters （fracturing pore size and peak pressure of detonation）on the

effect of liquid CO 2 phase change cracking.

I agree to accept this paper.

Reviewer #3: The author modifies the relevant problems in this paper. The research on influencing factors of the liquid CO2 phase change fracturing effect in coal seams is expressed comprehensively and accurately. It is proposed to be published.

Reviewer #4: (No Response)

7. PLOS authors have the option to publish the peer review history of their article (what does this mean?). If published, this will include your full peer review and any attached files.

Reviewer #1: No

Reviewer #3: No

Reviewer #4: No

---

## [Editor Report · Acceptance letter]

12 Jul 2021

PONE-D-21-16377R1 

Study of the influencing factors of the liquid CO_2_ phase change fracturing effect in coal seams 

Dear Dr. Li:

I'm pleased to inform you that your manuscript has been deemed suitable for publication in PLOS ONE. Congratulations! Your manuscript is now with our production department. 

Kind regards, 

on behalf of

Dr. Ahmed Mancy Mosa 

Academic Editor

PLOS ONE